# KHGCN: Knowledge-Enhanced Recommendation with Hierarchical Graph Capsule Network

**DOI:** 10.3390/e25040697

**Published:** 2023-04-20

**Authors:** Fukun Chen, Guisheng Yin, Yuxin Dong, Gesu Li, Weiqi Zhang

**Affiliations:** School of Computer Science and Technology, Harbin Engineering University, Harbin 150001, China

**Keywords:** knowledge graph, recommendation system, graph neural network, attention mechanism

## Abstract

Knowledge graphs as external information has become one of the mainstream directions of current recommendation systems. Various knowledge-graph-representation methods have been proposed to promote the development of knowledge graphs in related fields. Knowledge-graph-embedding methods can learn entity information and complex relationships between the entities in knowledge graphs. Furthermore, recently proposed graph neural networks can learn higher-order representations of entities and relationships in knowledge graphs. Therefore, the complete presentation in the knowledge graph enriches the item information and alleviates the cold start of the recommendation process and too-sparse data. However, the knowledge graph’s entire entity and relation representation in personalized recommendation tasks will introduce unnecessary noise information for different users. To learn the entity-relationship presentation in the knowledge graph while effectively removing noise information, we innovatively propose a model named knowledge—enhanced hierarchical graph capsule network (KHGCN), which can extract node embeddings in graphs while learning the hierarchical structure of graphs. Our model eliminates noisy entities and relationship representations in the knowledge graph by the entity disentangling for the recommendation and introduces the attentive mechanism to strengthen the knowledge-graph aggregation. Our model learns the presentation of entity relationships by an original graph capsule network. The capsule neural networks represent the structured information between the entities more completely. We validate the proposed model on real-world datasets, and the validation results demonstrate the model’s effectiveness.

## 1. Introduction

Recommendation systems have shown great potential in solving various online applications’ rapidly growing information volume problems, improving user efficiency, and increasing stickiness. In commercial platforms, such as Netflix and Amazon, recommendation systems provide excellent convenience for users by filtering out target items from thousands of movies and TV shows or millions of products [1,2]. To cope with the problems caused by the ever-increasing data, researchers have also developed neural network models and deep learning methods that process massive amounts of information. Deep neural networks have considerable advantages in many information-dominated fields, such as healthcare [3] and the Internet of Things (IoT) [4], especially in recommendation systems that rely on data [5,6].

Compared with traditional machine learning methods, deep learning methods can more effectively incorporate additional information (such as text, pictures, and so on) related to the recommendation task. Data-sparse and cold-start problems in recommendation systems (such as collaborative filtering-based recommendation systems) only utilizing user–item interactions for recommendation usually face a sharp drop in recommendation performance during user–user or user–item interactions. To alleviate the troubling data-sparse and cold-start problems in collaborative filtering (CF) based recommendation systems, researchers usually add the side information of users and items and design sophisticated algorithms to utilize this information. Various types of auxiliary information have been utilized to alleviate these problems.

Researchers use valuable external knowledge as extra information, such as reviews, social networks, tags, item attributes, etc., to improve the effectiveness of recommendations. Wang et al. used neural networks to extract the embedding of ratings and reviews, respectively. Further, they combined the embedding with a collaborative filtering method to propose a hybrid deep collaborative filtering model [7]. Chen et al. introduced label information into the recommendation system and proposed a label intersection model by studying the intersections between user labels and item labels for a better recommendation [8]. Shi et al. were motivated to utilize reviews and further reduced the dual graph convolutional network method to capture the full description of an aspect in all reviews for the recommendation [9]. The knowledge graph (KG) can encode users, items, and attributes related to items in the graph structure to preserve relationship information, thus attracting a wide range of research interests. Recently, researchers have explored the recommendation system based on the knowledge graph [10,11,12,13,14]. Knowledge-graph-embedding (KGE) methods integrate the KG at the recommendation system, knowledge-aware recommendation, to advance the accuracy and interpretability of the recommendation task, which has catalyzed considerable research works [15,16,17,18]. Researchers consider KG effective for improving quality recommendations because user and item attributes in KG are essential auxiliary information. Integrated interactions between the user and item and the attributes in KG (that is, they appear in the sample data simultaneously) can significantly improve the prediction accuracy in various recommendation systems.

Figure 1 depicts an example of a collaboration knowledge graph (CKG) in the film domain [19] to infer the preference of users $u_{4}$ and $u_{5}$ for item $i_{5}$. As shown in the user–item interactions section in Figure 1, the CF-based method determines whether to recommend or not by calculating the value of the degree of association (cosine similarity, etc.) among the $u_{4}$, $u_{5}$, and $i_{5}$. For example, for sim($u_{4},i_{5}$) = sim($u_{5},i_{5}$) = 0, the result is negative. In the knowledge graph part of Figure 1, item $i_{3}$ (preferred by user $u_{4}$), item $i_{4}$ (preferred by user $u_{5}$), and item $i_{5}$ have the same attribute $e_{3}$. We refer to the hybrid structure of the knowledge graph and user–item graph as CKG, which takes the user–item interactions and knowledge graph into account. According to the attribute $e_{3}$ in CKG, there is the interaction between users $u_{4}$, $u_{5}$, and movie $i_{5}$, which is connected by relation Gener. Obviously, at this time, the recommendation system can mine the user’s more prolonged, deeper, and more profound preference items to enhance the recommendation performance and interpretability of the recommendation system. Next, the latest GNN-based algorithms can further mine higher-order relationships between the user–item entities in CKG. However, to exploit higher-order information in the CKG graph, the following questions are noteworthy:With the increase in high-order size, the number of nodes related to the target user increases sharply, which increases the computational load of the model.Although the number of nodes related to target users has increased, nodes under higher-order relationships have different effects on the recommendation. Therefore, the contribution of nodes requires further screening by algorithmic models.The increase in nodes is a double-edged sword. Knowledge graphs inevitably introduce specific noise. Therefore, it is difficult for previous research methods to characterize target users and candidate items to generate an accurate recommendation list. Accurate user-embedding learning is essential for modern recommendation systems.

Graph structure information provides valuable guidance for graph neural networks learning node representation [20,21]. The excellent potential of the graph convolutional network (GCN) for recommending each is due to its ability to learn better embeddings of the user and item by using collaborative representations from high-level neighbor nodes [22,23]. Usually, attributes are not isolated but interconnected, which forms a KG. One must go beyond the obligation to model user–item interactions and consider auxiliary information to provide more accurate, diverse, and interpretable recommendations. Recently, leveraging knowledge graphs in recommendation systems to alleviate data-sparse and cold-start problems has attracted considerable attention. Despite recent advances in graph neural networks (GNNs), high-level collaboration signals are combined to alleviate the problem. Still, the ending of cold-start users and items is not optimized, and cold-start neighbors are not handled during startup. Solving the cold-start problem in recommendation systems is crucial for new users and new items. Under the premise of sparsely observed data, how to further mine unobserved user–item pairs is also an important research direction to refine users’ potential preferences. The other social relationships are usually used to improve the recommendation quality when considering the sparsity of the user–item interactions in the social recommendation.

Graph neural networks (GNNs) have significant advantages in the display modeling of structured data. However, existing GNNs are limited in their ability to capture the representation of the hierarchy, and the hierarchy graph plays an important role in the representation learning of the graph. Like other GNN models, GNN-based recommendation models inevitably suffer from the problem of over-smoothing. Over-smoothing is when the graph neural network stacks more layers, and the node embeddings in the network become increasingly similar until they become indistinguishable, resulting in performance degradation. Aiming at the problems of graph representation learning, researchers try to use the multi-channel feature of capsGNN to learn a complete graph structure from the perspective of breadth [24,25,26,27,28]. In graph representation learning, capsule GNN has also achieved good results [29]. Xu et al. proposed a taxonomy-enhanced graph neural network (Taxo-GNN), which jointly optimizes the taxonomy representation and node representation tasks, where categories in taxonomy are mapped to Gaussian distributions and nodes are embedded with the GNN framework [30]. Inspired by the capsule neural network (CapsNet) [31], we propose a hierarchical graph capsule network (HGCN), which uses the capsule concept to solve the shortcomings of existing GNN-based graph-embedding algorithms. By extracting node features in the form of capsules, the routing mechanism can be used to capture important information in the hierarchy graph. However, there are noisy nodes in the graph structure constructed by the knowledge graph. Most GNN-based recommendations will also learn the representation of the noisy node when learning the presentation of graph structure information. To alleviate the restrictions of the recommendation methods with KG and GNN, we propose a novel transformer graph attention network (TGAT) component in our model for high-order information propagation in the collaborative knowledge graph. The attentive mechanism in TGAT is weighted to judge the importance of each entity; the attention mechanism is used to extract the significance of each channel; and multiple screenings weaken the significance of noisy entities and strengthen the reputation of crucial entities. We summarize the problems existing in the recommendation algorithms with KG and GNN. From the perspective of a knowledge graph, there are noisy entities and relationships in a complete knowledge graph. From the perspective of graph neural network methods, graph neural networks, such as GCN itself, face the problem of model bottlenecks. To solve the problems that CKG and graph neural networks face in recommendation systems, we propose the knowledge-enhanced hierarchical graph capsule network method, which alleviates the problems existing in the incorrect forms. In general, the contribution of our model is as follows:Node disentangle is introduced, and the problem of the representation of noisy entities and relationships in the complete KG for recommendations is alleviated, generating a disentangled user–item knowledge graph;A novel attention scoring function is designed to more effectively aggregate the nodes in the disentangled knowledge graph and create a more accurate representation of the user and item;The introduction of KHGCN, the use of the hierarchical transformer graph attention network (TGAT) to enhance the ability to represent relationships and entities in the graph, learn the representation of entity relationships in the KG more efficiently, and make accurate recommendations;An end-to-end model recommendation framework is constructed, surpassing existing state-of-the-art methods on three real-world datasets and four evaluation indicators.

## 2. Related Work

### 2.1. Knowledge Graph for Recommendations

Introducing the knowledge graph for the recommendation system can effectively alleviate the data-sparse and cold-start problems. In recent years, many researchers have been performing related work. To model both the dynamic interests and the dynamic social influences, Gu et al. proposed a method to model and integrate item-embedding representations and contextual friendship representations for recommendation tasks [32].

To integrate the knowledge graph with the recommendation system under a unified framework, researchers combine the knowledge graph’s feature learning with the recommendation algorithm’s objective function and train in the same end-to-end objective process. Zhang et al. exploited heterogeneous information in the knowledge graph to enhance the performance of recommendation systems. They proposed that the integrated framework of collaborative knowledge-graph embedding is learning semantic representations in items from the knowledge graph while learning latent representations of methods in collaborative filtering [33]. Dong et al. proposed a novel and simple model to achieve a low computational cost, which adopts a semi-automatic encoder to embed item attributes and graph features simultaneously for recommendations [34]. In [12], a hybrid recommendation system based on attention mechanism and knowledge-graph embedding (HRS) is put forward to alleviate sparsity and cold-start problems and improve the performance of the recommender systems. Zou et al. focused on exploring the contrastive learning in KG-aware recommendation and proposed a novel multi-level cross-view contrastive learning mechanism (MCCLK), which comprehensively considers three different graph views for KG-aware recommendation, including the global-level structural view, local-level collaborative and semantic views [10]. In [14], Zou et al. focused on exploring contrastive learning in KGR and proposed a novel multi-level interactive contrastive learning mechanism, which contrasts nodes of two generated graph views. To tackle the problem that most existing KG-based recommender systems ignore the fact that users attach different degrees of importance to various relationships of items, Zhang et al. proposed a knowledge-graph recommender model based on adaptive relational attention (KGARA) [11]. Yin et al. proposed a model for knowledge-aware recommendation systems, where the main components are an attribute regularizer and a dynamic attention mechanism, which are attention components. They explored potential connections between the users and items to enhance the recommendation performance [35].

Since both recommendation and knowledge-graph embedding tasks are included, knowledge-based recommendation systems are similar to multi-task learning frameworks, which are related by items and entities in the knowledge graph. Knowledge-graph representation and recommendation tasks are inevitably related. The knowledge-graph link prediction can assist the recommendation system to keep away from the local minimum and prevent the recommendation system from overfitting, thereby improving generalization. Some researchers regard recommendation systems and knowledge-graph feature learning separately and only adopt a multi-task learning framework in the final loss function training. Wang et al. proposed a compression unit to correlate the two tasks, which automatically shares latent features while also learning higher-order interactions between the user–item interactions and knowledge-graph entities [36]. The structured data features of knowledge graphs have unique advantages compared with other auxiliary information in personalized recommendation systems, which enhance the interpretability of the recommendations’ knowledge-based representation learning framework to embed heterogeneous entities for the recommendation based on the knowledge-graph embedding, and a soft matching algorithm is proposed. Regarding personalized explanations for the recommended items [37], Liu et al. proposed a knowledge-graph-enhanced multi-task learning method to learn cross features from ratings and reviews by fusing users and movies with their review knowledge entities in the same graph [13].

### 2.2. GNN for Recommendations

With the deepening of knowledge-based recommendation-systems research, the relational structure of external knowledge can be introduced while learning user and item representations. In this regard, traditional neural networks have successfully extracted features from Euclidean spatial data but, in many cases, still contain a lot of non-Euclidean spatial data. However, the performance of traditional neural networks is still unsatisfactory when addressing non-Euclidean spatial data. As the performance is still unsatisfactory, the researchers draw on the ideas of convolutional neural networks, recurrent neural networks, and deep auto-encoders to define the graph neural networks that can process non-Euclidean spatial data. Since then, the related research of graph neural networks has sprung up as new hot research.

In recent years, researchers have used GNN for recommendations with great success. Thomas N. Kipf and Max Welling proposed a graph convolutional new algorithm derived from the idea of the convolution algorithm and able to be directly used to process structural graph data [20]. Another feature of GCN is that its model size increases linearly with the number of edges in the graph. Given the problems of GCN addressing large-scale graphs, GraphSAGE is proposed for inductive representation on large graphs. GraphSAGE is a framework for inductive representation learning on large graphs, which is used to generate low-dimensional vector representations of nodes that can efficiently process graphs with many nodes’ attribute information [20]. For the fine-grained division of node attributes information in a graph, Petar et al. proposed new neural network architecture, the graph attention network (GAT), which learns the node weights in the graph structure through a masked self-attention layer [21]. It effectively alleviates the shortcomings of other methods, such as GCN.

Some researchers have studied recommendations based on graph neural networks, among which the works based on the GCN framework are as follows. In [38], Chen et al. proposed a GCN-based linear collaborative model with a residual network component, which can relieve the over-smoothing problem in higher-order graph convolution to a certain extent. Furthermore, He et al., through empirical observation, found that the two components of feature transformation and nonlinear activation in the structure of GCNs do not contribute much to improving collaborative filtering. A new model named LightGCN based on GCN is proposed, and the fundamental neighbor aggregation part in GCN is reserved for collaborative filtering [23]. In [39], the problem of hashing with GNN for high-quality retrieval was investigated, and a deep hashing with GNN framework was proposed to learn continuous and GNN codes jointly. Wu et al. proposed an adaptive GCN method for knowledge-based recommendations, effectively integrating the item recommendations and attribute inference [40]. The attribute inference part in adaptive GCN can adjust the graph-embedding learning parameters and refine the item attributes and preference behavior through the weak supervision information provided by the attribute inference part to complete the item recommendation. Previous studies failed to model the various impacts of multi-behavior in multi-behavior recommendations. In [41], a new model, multi-behavior GCN, was proposed to solve the problems in multi-action recommendation tasks. The main idea of multi-behavior GCN is to construct a multi-behavior graph and then learn the representation of the multi-behavior graph through an improved graph neural network. To solve food-recommendation problems, Gao et al. proposed an innovative graph convolutional network, which hierarchically learns the relationships in the food graph and models higher-order relationships in the food graph through an information propagation mechanism [42]. To solve oscillation problems, Liu et al. proposed a new model for recommendation tasks, named deoscillated adaptive graph collaborative filtering, which includes the layer-wise propagation patterns and can adaptively learn local factors [43]. To learn the fine-grained property features, Ge et al. proposed an end-to-end model called the collaborative property-aware graph convolutional network (CPGCN), which fuses the user–service collaborative information with the semantic information of KG to construct collaborative property-aware graphs (CPGs) [44]. Wang et al. proposed a new knowledge graph-aware light graph convolutional network (KLGCN), which removes feature transformation and nonlinear activation in KG-aware recommendation while improving performance [45]. Zhu et al. embedded users’ interests from their social information by the attentional graph convolutional network (GCN) and improved news representations via attention mechanisms [46].

The works based on the GraphSage framework are as follows. In [47], a matrix completion model based on the inductive graph is proposed, which performs 1-hop subgraph expansion through the user–item interaction graph generated based on the rating matrix, and aggregates these subgraph representations through GNN. Sun et al. proposed a new framework that can explicitly aggregate neighbor node information, efficiently modeling user–item bipartite heterogeneous graphs [48]. In the online recommendation services, Xu et al. developed a framework based on incremental learning graph neural networks to address the catastrophic forgetting problem faced during incremental learning training, which implements a graph structure preservation strategy to preserve users’ long-term preferences when the model is updated [49]. In the personalized video highlight recommendations, Wu et al. proposed an inductive transfer learning framework consisting of a graph neural network and an item embedding transfer network for the personalized recommendations of video highlights [50].

The works based on the GAT framework are as follows. To capture fine-grained user preference, in [51], an inductive transfer learning framework was proposed, which consists of a graph neural network and an item embedding transfer network for the personalized recommendations of video highlights. Regarding social recommendations, Wu et al. proposed a method (DiffNet++) based on the DiffNet to combine social networks and interest networks in social recommendations to construct a new social interest heterogeneous graph to redefine social recommendations [52]. Furthermore, DiffNet++ injects user–user high-order implicit feedback in the social network, injects user–item high-order implicit feedback in the interest network, and infiltrates the user-embedding representation in a fine-grained manner. To effectively learn the contribution of different neighbor entities to the target embedded in the process of neighborhood propagation, they developed a hierarchical attention mechanism at the neighbor level and graph level, which can adaptively learn important content at different levels, and provides more possibilities for social recommendations  [53]. In multimodal recommendations, Sun et al. proposed multi-modal knowledge GAT, a multimodal graph attention technology that uses a graph attention network to model multimodal information on a multimodal knowledge graph [54]. The modal knowledge-graph attention mechanism integrates the text and image information of entities to the tail node of the knowledge graph, which enriches the entity representation of the knowledge graph. Feng et al. proposed a model that can effectively capture user–item pairs in knowledge graphs and employ graph-connection and graph-pruning techniques to construct the behavioral graphs of adaptive targets in collaborative knowledge graphs [55]. Wang et al. proposed a new method called knowledge GAT (KGAT), which innovatively handles knowledge graphs with graph attention neural networks and proposes a new neighbor aggregation method to improve the target node’s embedding; the attention mechanism in KGAT can effectively divide the importance of neighbor nodes [19].

## 3. Models and Methods of KHGCN for Recommendation

This section discusses in detail the knowledge-enhanced hierarchical graph capsule network (KHGCN) proposed to address the problems of knowledge-based recommendation tasks. KHGCN aims to learn node embeddings in collaborative knowledge graphs as well as fine-grained learning of the representation of target nodes in the CKG using a hierarchical graph capsule neural network. We take the dissociated representation of nodes in a collaborative knowledge graph as input to a graph capsule neural network. Therefore, the primary capsule in each graph capsule neural network is composed of multiple disentangled independent latent factors, where each latent factor represents a different attribute of the entity. Transformer GATs (TGATs) generate the primary capsules, whose main function is to encode the part–whole relationship between the low level and high level by further integrating the high-order neighbor representations of target nodes in collaborative knowledge graphs. Specifically, the instantiation parameters in the upper-layer capsules are obtained by the voting of the lower-layer capsules TGAT, which depends on the structure between the lower-layer capsules. The lower layer is routed to high-level capsules through the routing mechanism of the capsule neural networks. After several rounds of iterations, high-order representations of user items are obtained through objective function training. Finally, the obtained user and item representations are fitted by the prediction layer to complete the recommendation. Figure 2 shows the description of the KHGCN model framework and other components.

### 3.1. Problem Formulation

In this section, we introduce a detailed definition of the KHGCN model for the CKG-based recommendation. According to the collaborative knowledge graph constructed in Figure 1, the datasets are divided into the following three parts for introduction. User–item interaction graph: This contains users, items, and user–item interactions. The set of users is denoted as $U=\left\{u_{1},u_{2},\cdots ,u_{\left|U\right|}\right\}$. The set of items is denoted as $V=\{v_{1},v_{2},\cdots ,v_{\left|V\right|}\}$. Moreover, the user–item interactions set is denoted as $Y=\{y_{uv}|u\in U,v\in V\}$. For each interaction, $y_{uv}=1$ means that the user *u* has positive feedback for item *v*, and $y_{uv}=0$ means that the user *u* has no feedback for item *v*.

Knowledge graph: The knowledge graph consists of entity connection relationships. The entity set is denoted as *E* = $\{e_{1}$, $e_{2}$, ⋯, $e_{\left|E\right|}\}$, and |E| represents the number of entities in knowledge graph. The relationship set is denoted as *R* = $\{r_{1}$, $r_{2}$, ⋯, $r_{\left|R\right|}\}$, and |R| represents the number of relations in knowledge graph. where (h,r,t) is a triple in G, and *h*, *t* represent the entities of head and tail, and relation *r* represents the relation between the head *h* and tail *t*. Here, we adopt a set of triples to represent the entire knowledge graph G(E,R,E).

Collaborative knowledge graph: This is a joint user–item graph and knowledge graph, for which we define user–item interactions as ${\hat{y}}_{uv}=ϝ\left(u;v;\Theta ,Y,G\right)$, where ${\hat{y}}_{uv}$ represents the predicted interaction between the user *u* and item *v* pairs in the collaborative knowledge graph, and Θ is the parameter of the function ϝ.

Candidate items set for the target user are generated according to the predicted interactions. Some of them are selected for the recommendation list according to the actual situation. The function ϝ is defined by the KHGCN. KHGCN generates the top@*N* list for the CKG-based recommendation, where *N* is the number of items in the recommendation list.

Figure 2a shows the overall framework of the recommendation model KHGCN, where the main components of the model are the embedding of CKG, disentanglement, capsule neural networks, and hierarchical graph attention networks.

### 3.2. Knowledge Graph Embedding

Knowledge-graph embedding maps entities and relations in a knowledge graph to a unified space for efficient representation, while preserving the structure of the knowledge graph. KHGCN adopts TransR [17], a widely used KGE method, to learn representations of knowledge graphs. Specifically, TransR learns the translation principle for (h,r,t) triples and learns to embed each triple existing in the knowledge graph by optimizing the translation principle $e_{h}^{r}+e_{r}\approx e_{t}^{r}$. Where $e_{h},e_{t}\in R^{d}$ and $e_{r}\in R^{k}$, *d* are the embedding dimensions of the entity in the knowledge graph, *k* is the embedding dimension of the relation in the knowledge graph, and $e_{h}^{r}$ and $e_{t}^{r}$ are the mappings of the entities *h* and *t* in the relation *r* space. To sum up, given the triples present in the knowledge graph, the energy score formula calculated by TransR is as follows: (1)$$g\left(h,r,t\right)=\left|\right|W_{r}e_{h}+e_{r}-W_{r}e_{t}{\left|\right|}_{2}^{2}$$
where $W_{r}\in R^{k\times d}$ is the spatial transformation matrix of relation *r*, and the transformation matrix is used to map *d*- dimensional entities into *k*-dimensional relation space *r*. In the formula, if the triplet has a higher score, the triplet is likely to be not true, and the lower the triplet score, the more likely it is to be true.

### 3.3. Node Disentangle

In most cases, highly complex interactions are involved in connecting each node pair in a graph. For example, the researchers largely model the relationships uniformly as before while neglecting the diversity of user intentions in watching the films, which could be for the director, actors, accompanying family or friends, etc. Therefore, it is necessary to disentangle the interpretable latent factors underlying the variation of entity representations in knowledge graphs. We expect to completely dissociate the independence between different intents in entity representations [29,43].

Motivated by [31], we add the disentanglement component into our model, which can be called disentangled HGCN, to disentangle the entities’ representation factors and focus on disentangled representations for recommendation tasks. The dissociated entity node is used as the input of the graph capsule. Each graph capsule is composed of multiple independent heterogeneous factors, where each factor describes a piece of the representation of the entity.

For a knowledge graph, G=(E,R,E), node *i* in the graph is represented by $e_{i}\in R^{d}$. Specifically, we project the input entity features into *K* different subspaces, assuming there are *K* latent factor parameters: (2)$$e_{i,k}=\sigma \left(W_{k}^{T}e_{i}\right)+b_{k}$$
where $W_{k}\in R^{d\times \frac{d_{h}}{K}}$, and $b_{k}\in R^{\frac{d_{h}}{K}}$ are learnable parameters, σ is a nonlinear activation function, and $\frac{d_{h}}{K}$ is the dimension of each factor. Our study uses linear projection due to its efficiency and remarkable performance. Therefore, each graph capsule is represented by a pose matrix $E_{i}\in R^{K\times \frac{d_{h}}{K}}$[56].

After the dissociation operation, we reshape $E_{i}$ into the dissociated vector format $E_{i}\in R^{d_{h}}$. At this point, we obtain the de-dissociated entity vector to match the capsule graph’s input. Therefore, we compress $z_{i}$ as follows [31]: (3)$$p_{i}=squash\left(e_{i}\right)=\frac{\left|\right|e_{i}{\left|\right|}^{2}}{1+\left|\right|e_{i}{\left|\right|}^{2}}\frac{e_{i}}{\left|\right|e_{i}\left|\right|}$$
where $e_{i}^{\left(1\right)}=p_{i}\in R^{d_{h}}$ is the primary graph capsule representing the lowest-level entity and the basis upon which all advanced capsules are founded.

### 3.4. Hierarchical Graph Capsule Learning

We propose the hierarchical graph capsule layers, consisting of TGATs and residual routing parts. The task of TGATs is to vote for the instantiation parameters of higher-level graph capsules.

Figure 2c shows the attentive embedding propagation layer part. Unlike KGAT [19], we pass the dissociated vector through a layer of the self-attention mechanism with residual values and then feed the resulting vector into the improved attention mechanism and embed the propagation layer.

#### 3.4.1. Transformer Graph Attention Layers

ResidualTransformer: As shown in Figure 2b, we add a residual self-attention mechanism before the attention embedding propagation, which facilitates better coarse extraction of the embedding representation of the graph:
(4)$$f_{RT}\left(X\right)=\left(\lambda_{t}+softmax\left(\frac{Q_{t}{K_{t}}^{T}}{\sqrt{d_{K_{t}}}}\right)V_{t}\right)X$$
where $Q_{t}$ = $K_{t}$ = $V_{t}$ = *X*, $\lambda_{t}$ = 1, and *X* = $e_{i}^{\left(1\right)}$ is the representation of disentangled entity embedding and relation in knowledge-enhanced graph *G*. The embedding of the *i*-th node $e_{i}^{\left(1\right)}$ can be represented as $e_{i}^{\left(1\right)}$= $f_{RT}\left(e_{i}^{\left(1\right)}\right)$, where RT is short for residual-transformer.Knowledge−awaregraphattention: For each entity *h* in graph *G*, we define its associated neighbor entity as $N_{h}=(h,r,t)|(h,r,t)\in G$, where $N_{h}$ represents the set of triples associated with the head node *h*. The attentive embedding propagation layer part is shown in Figure 2d.To characterize the first-order connectivity structure of entity *h*, we obtain the relevant first-order neighbor embedding by computing the relation score π and the relevant entity *t* in $N_{h}$ as
(5)$$e_{N_{h}}=\sum_{\left(h,r,t\right)\in \left(N_{h}\right)}\pi \left(h,r,t\right)e_{t}$$
where π(h,r,t) controls the decay factor on each propagation on edge (h,r,t), indicating how much information is being propagated from *t* to *h* conditioned to relation *r*. In this part, we construct a new formula, and the learned node embedding with attention is integrated with the bi-interaction function. The improved formula for calculating π(h,r,t) is as follows:
(6)$$\pi \left(h,r,t\right)={\left(\tanh\left(W_{r}e_{t}+e_{r}\right)\right)}^{T}\tanh\left(W_{r}e_{h}+e_{r}\right)$$
where we select tanh as the nonlinear activation function. The attention score π(h,r,t) between nodes *h* and *t* is determined by the relational spatial distance of relation *r* in TransR. The higher the degree of association to node *h* in the relationship space, the higher the value of the attention score π(h,r,t). After calculating the scores of all nodes *t* related to node *h* in $N_{h}$, we normalize the attention scores by the softmax function:
(7)$$\pi \left(h,r,t\right)=\frac{exp(\pi (h,r,t)}{\sum_{\left(h,r^{\prime },t^{\prime }\right)\in N_{h}}exp(\pi \left(h,r^{\prime },t^{\prime }\right))}$$
Therefore, the final attention score determines that nodes with high scores should be given more weight and attention so that high-order features in the neighbor set of the target node can be learned more accurately. During the training process, the attention scores divide the importance of each datum while enhancing the interpretability of the recommendation.Bi−interactionaggregator: The Bi-interaction aggregator is the result of considering various feature interactions between $e_{h}$ and $e_{N_{h}}$. The Bi-interaction calculation formula is as follows:
(8)$$f_{Bi}=LeakyRelu(W_{1}\left(e_{h}+e_{N_{h}}\right)+LeakyRelu\left(W_{2}\left(e_{h}⊙e_{N_{h}}\right)\right)$$
where $W_{1}$, and $W_{2}$∈$R^{d^{\prime }\times d_{h}}$ are the trainable weight matrices, and ⊙ denotes the element-wise product. We encode the feature interactions between $e_{h}$ and $e_{N_{h}}$. This term makes the information being propagated sensitive to the affinity between $e_{h}$ and $e_{N_{h}}$, e.g., passing more messages from similar entities, where Bi is short for Bi-interaction. We encode the target node *h* embedding and its corresponding neighbors’ set $N_{h}$ embedding by Bi. Through the encoding method of Bi, we can further encode the high-order connection of the target node in the graph. Our method is built upon TGATs by following the attention information propagation paradigm:
(9)$$e_{h}^{\left(l+1\right)}=f_{Bi}\left(e_{h}^{\left(l\right)},e_{N_{h}}^{\left(l\right)}\right)$$
The final representation of node $e_{h}$ is given by TGAT($e_{h}^{\left(1\right)}$, $e_{N_{h}}^{\left(1\right)}$) = $e_{h}^{\left(L\right)}$∈$R^{L\times d_{h}}$ after *L* iterations.

#### 3.4.2. Learning Primary Capsules

For all primary capsules in layer *l*, generated by the proposed hierarchical TGATs, it is assumed that the *m*-th node embedding of layer *l* can be calculated via
(10)$$p_{j}^{\left(l\right)}=TGAT_{j}\left(e_{i}^{\left(l\right)},e_{N_{i}}^{\left(l\right)}\right)$$

After stacking the hierarchical TAGTs outputs, we concatenate the primary capsules of node *i* at layer *l* into a vector $p^{\left(l\right)}$. The *j*-th primary capsule is represented as $p_{j}^{\left(l\right)}$, where $p_{j}^{\left(l\right)}\in R^{L\times d_{h}}$.

#### 3.4.3. Routing for Graph Capsules

After the primary capsule is generated, the obtained primary capsule generates the graph capsule through the graph routing mechanism. To iteratively generate the graph capsules, in each iteration, there are
(11)$$e_{i}^{\left(l+1\right)}=squash\left(\sum_{i}c_{i,j}^{\left(l\right)}p_{j|i}^{\left(l\right)}\right)$$
where $c_{i,j}^{\left(l\right)}\geq 0$, and $\sum_{i=1}^{L}c_{i,j}^{\left(l\right)}=1$. $q_{j}^{\left(l+1\right)}$ is the *j*-th graph capsule in layer l+1, representing the close voting cluster from the primary capsules in layer l+1, and $c_{i,j}^{\left(l+1\right)}$ is the routing coefficient used to calculate the voting of each primary capsule for the graph capsule, which represents the primary capsule $p_{j}^{\left(l\right)}$ relative to the graph capsule $e^{\left(l+1\right)}$.Here, consider the importance of $c_{i,j}^{\left(l\right)}$, which is iteratively updated using a graph routing mechanism.

The vote of each capsule in the primary capsules $p^{\left(l\right)}$ is routed to a capsule in the graph capsule $e^{\left(l+1\right)}$, and each graph capsule has the participation of the primary capsule. Formally, the vote routing coefficient $c_{i,j}^{\left(l\right)}$ is defined by softmax:(12)$$c_{i,j}^{\left(l\right)}=exp\left(b_{i,j}^{\left(l\right)}\right)/\sum_{s}exp\left(b_{i,s}^{\left(l\right)}\right)$$
where the initial value of $b_{i,j}^{\left(l\right)}$ is defined as 0. We perform *R* iterations of the graph-routing mechanism at each iteration, and $b_{i,j}^{\left(l\right)}$ is also updated as follows: (13)$$\begin{array}{cc} b_{i,j}^{\left(l\right)} & =b_{i,j}^{\left(l\right)}+a_{i,j}^{\left(l\right)},\\ a_{i,j}^{\left(l\right)} & =p_{j|i}^{\left(l\right)}{\dot{q}}_{j}^{\left(l+1\right)} \end{array}$$
where $a_{i,j}^{\left(l\right)}$ indicates the agreement between each vote and vote cluster. After *R* iterations, we obtain higher-level graph capsules $e^{\left(l+1\right)}$. Figure 2b shows that we add a residual connection to each pair of consecutive graph capsule layers to provide richer information for higher-level graph capsules.

To reduce the number of trainable parameters in KHGCN, we set $d_{h}$ = *d* in our experiments. It has been proven that such a reduction does not affect the final recommendation performance. Formally, the output of the graph capsule in layer (l+1) is defined as $e^{\left(l+1\right)}\leftarrow e^{\left(l+1\right)}+GA\left(e^{\left(l\right)}\right)$, where GA indicates the global average operation.

### 3.5. Model Prediction

After hierarchical graph capsules perform *L*-order information propagation, we obtain all representations of user node *u* at the *L*-level in the graph, namely $\{e_{u}^{\left(1\right)}$, …, $e_{u}^{\left(L\right)}\}$. Likewise, we obtain the representation of item node *v*, namely $\{e_{v}^{\left(1\right)}$, …, $e_{v}^{\left(L\right)}\}$. Therefore, we adopt the general layer-aggregation method to concatenate the representations of each layer into a vector to represent the target node via
(14)$$e_{u}^{⋇}=[e_{u}^{\left(0\right)}\left||\ldots |\right|e_{u}^{\left(L\right)}],e_{v}^{⋇}=[e_{v}^{\left(0\right)}\left||\ldots |\right|e_{v}^{\left(L\right)}]$$
where || is the concatenate mechanism for vectors.

We enrich the initial embedding by the embedding propagation operation and control the strength of information propagation by adjusting the parameter *L* simultaneously. In the prediction layer, we use an inner product operation on the vector of the user and item to predict the interaction probability between them via
(15)$$\hat{y}\left(u,v\right)=e_{u}^{⋇T}e_{v}^{⋇}$$

### 3.6. Loss Function and Model Training

Training for TransR considers the relative order of positive and negative triples pairwise and trains with the pairwise ranking loss for overall ordering via
(16)$$L_{KG}=\sum_{h,r,t,t^{\prime }\in F}-ln\sigma \left(g\left(h,r,t^{\prime }\right)-g\left(h,r,t\right)\right)$$
where $F=\left(h,r,t,t^{\prime }\right)|\left(h,r,t\right)\in G,\left(h,r,t^{\prime }\right)\notin G$, and  $\left(h,r,t^{\prime }\right)$ is the negative triple constructed by randomly replacing one entity in the positive triple, and $\sigma \left(\cdot \right)$ is the sigmoid function.

This layer models the entities and relationships through the graph triples. The knowledge graph loss function $L_{KG}$ is used as a regularization item for the auxiliary recommendation to train together with the recommendation model, thereby improving the generalization ability of the recommendation model.

Likewise, our training construct for the recommendation part comes after obtaining the output ${\hat{y}}_{uv}$ of the prediction layer. In the prediction layer, the likelihood of user *u* and item *v* is estimated by feeding their final representations into the prediction function. We also use pairwise BPR loss to optimize the recommended partial model parameters Θ={u,v|u∈U,v∈V}. Specifically, since generated user–item interactions are identified user preferences, it is assumed that identified interactions should be assigned higher predictive values than unidentified interactions:(17)$$L_{Rec}=\sum_{u,v,w\in O}-ln\sigma \left(\hat{y}\left(u,v\right)-\hat{y}\left(u,w\right)\right)$$
where $O=\{\left(u,v,w\right)|\left(u,v\right)\in R^{+},\left(u,w\right)\in R^{-}\}$ denotes the training set, $R^{+}$ denotes the observed (positive) interaction between the user *u* and item *v*, and $R^{-}$ is the sampled unobserved (negative) interactions set. $\sigma \left(\cdot \right)$ is the sigmoid function.

Finally, we have the objective function to learn Equation (Equation 16) and Equation (Equation 17) jointly, as follows: (18)$$L_{KHGCN}=L_{KG}+L_{Rec}+\lambda_{\Theta }{∥\Theta ∥}_{2}^{2}$$
where Θ = {E, $W_{r}$, ∀l∈R,$W_{1}^{\left(l\right)}$, $W_{2}^{\left(l\right)}$, ∀l∈{1,…,L}} is the model parameter set, E is the embedding representation of all entities and relations, and L2 regularization is performed on $\lambda_{\Theta }$ to prevent overfitting, where Θ is a parameter for controlling the regularization.

The whole training process is detailed in Algorithm 1. We optimize $L_{KG}$ and $L_{Rec}$ alternatively, and mini-batch Adam [57] is used to optimize the loss function $L_{KHGCN}$ of the collaborative knowledge graph recommendation model. Adam can adaptively control the absolute value of the learning rate with respect to the gradient and is an ordinary universal optimizer in deep learning.
**Algorithm 1** Procedure of KHGCN.**Input:** users *u*; items *v*; Interaction matrix *Y*; knowledge graph G**Output:** recommend the top@*N* item list  1:initialize all parameters, shuffle (*u*,*v*,*Y*,G);  2:**for** i=1 to |U|+|V|+|E| **do**  3:    apply TransR to obtain the embedding of the graph G  4:    **for** k=1 to *K* **do**  5:        $z_{i,k}=\sigma \left(W_{k}^{T}{\dot{x}}_{i}\right)+b_{k}$  6:    **end for**  7:**end for**  8:obtain the $L_{KGs}$ loss of the TransR from Equation (Equation 16);  9:$u_{i}^{\left(1\right)}=squash\left(z_{i}\right)$10:**for** l=1 to *L* **do**11:    $b_{i,j}^{\left(l\right)}=0$12:    **for** j=1 to $|N_{i}^{\left(l+1\right)}|$ **do**13:        $p_{j}^{\left(l\right)}=TGAT_{j}\left(e_{i}^{\left(l\right)},e_{N_{i}}^{\left(l\right)}\right)$14:    **end for**15:    **for** r=1 to *R* **do**16:        $c_{i,j}^{\left(l\right)}=exp\left(b_{i,j}^{\left(l\right)}\right)/\sum_{s}exp\left(b_{i,s}^{\left(l\right)}\right)$17:        $e_{j}^{\left(l+1\right)}=squash\left(\sum_{i}c_{i,j}^{\left(l\right)}p_{j|i}^{\left(l\right)}\right)$18:        $b_{i,j}^{\left(l\right)}=b_{i,j}^{\left(l\right)}+p_{j|i}^{\left(l\right)}\cdot e_{j}^{\left(l+1\right)}$19:    **end for**20:    $e_{i}^{\left(l+1\right)}$=GA($e_{1}^{\left(l+1\right)}$||,...,||$e_{j}^{\left(l+1\right)}$||,...,||$e_{|N_{i}^{\left(l+1\right)}|}^{\left(l+1\right)}$)21:    $e_{i}^{\left(l+1\right)}=e_{i}^{\left(l+1\right)}+GA\left(e_{i}^{\left(l\right)}\right)$22:**end for**23:Concatenate $\left(e_{u}^{\left(1\right)},\ldots ,e_{u}^{\left(L\right)}\right)$, Concatenate $\left(e_{v}^{\left(1\right)},\ldots ,e_{v}^{\left(L\right)}\right)$24:from prediction layer to calculate predicted probability ${\hat{y}}_{uv}$;25:compute the cross-entropy loss of $L_{Rec}$ from Equation (Equation 17);26:compute the total-loss $L_{KHGCN}$ from Equations (Equation 18);27:Apply Equations (Equation 16)–(Equation 18) to obtain the back-propagated loss error and update parameters through the entire network;28:update weights by the optimizer and update the learning rate *a*; **return** top@*N* item list;

## 4. Experiment

In this section, to evaluate the effectiveness of our model, we carry out experiments on three real-world benchmark datasets, Amazon-Book, Last-FM, and Yelp2018, which are publicly accessible. Furthermore, the three datasets are different in application fields, and there are some differences in data size and terms of data sparsity as shown in Table 1.

### 4.1. Dataset Description

Amazon−Book: Amazon-review is a widely used product recommendation dataset [58]. Amazon product data include some user–item data provided by Amazon. These data contain various Amazon products, such as books, electronics, movies and TV, home kitchens, outdoor sports, digital music, musical instruments, etc. This dataset includes reviews (ratings, text, and help votes), product metadata (description, category information, price, branding, and image features), and links (see/also buy charts). We choose Amazon-Book from Amazon for model performance evaluation, and we keep users and items with at least ten interactions (10-core).Last−FM: This is a dataset that provides music recommendations. Listening records of 92,800 singers from 1892 users; each user in the dataset contains a list of their most popular artists and the number of plays. It also includes user-applied labels that can be used to construct content vectors from the music listening dataset collected from the Last-FM online music system. Among them, tracks are considered items. In particular, we take a subset of the dataset with timestamps from January 2015 to June 2015. We utilize the same 10-core setup to ensure data quality.Yelp2018: This dataset was adopted from the 2018 edition of the Yelp challenge. This dataset covers business, review, and user data, and can be used for personal, educational, and academic purposes. Here, we consider local businesses such as restaurants and hotels as projects. Again, we use a 10-core setting to ensure at least ten interactions per user and item.To build a collaborative knowledge graph for recommendations, we introduce a knowledge graph based on the original user–item interactions in each dataset. As shown in Figure 1, the knowledge graph is introduced through the item side in our model. We first traverse each item in the Amazon book and LastFM datasets. If the mapping between the item and the knowledge graph is available (there is an intersection), we map the item to the Freebase entity through the title-matching method used in KGAT [19]. For Yelp2018, we extract item knowledge from local business information networks as KG data. To guarantee the integrity of the constructed knowledge-aware dataset, we consider all triples directly related to item-aligned entities. The statistics of the three knowledge-aware datasets are shown in Table 1 When constructing knowledge graph data, to guarantee the quality of the constructed knowledge graph, we filter the uncommon entities (that is, fewer than 10 in the intersection of both datasets) and retain at least 50 relations that appear in triples to preprocess the knowledge graph part of the three datasets. We randomly select 80% of each user’s interaction history for each dataset to form the training set to train the model parameters and the remaining 20% as the test set to verify the model performance. We randomly select 10% of each user’s interactions from the training set as the validation set to tune the model’s hyperparameters. We take each of its observed user–item interactions as a positive example and then pair it with a negative example with which the user has not interacted before through a negative sampling strategy.

### 4.2. Baseline Methods

FM [2]: FMs model all interactions between the inputs variables using factorized parameters. Here, we take the ids of the constructed collaborative knowledge graph as input features for FM.NFM [59]: This method is a state-of-the-art decomposition model that uses the properties of neural networks to fit arbitrary functions to highlight generalized NFMs, treating FMs as a particular case of NFMs.BPRMF [60]: Most algorithms are based on user predictions of product ratings for implicit feedback data. From the perspective of sorting, BPRMF sorts according to each user’s preference, in which the top-ranked items have higher priority.CKE [33]: Collaborative knowledge-base embedding (CKE) leverages the additional information in the knowledge base to improve the quality of recommendation systems and learns latent representations of entities related to items in collaborative filtering from the knowledge base.ECFKG [37]: The ECFKG model applies TransE [16] to the unified graph to embed relevant entities in the knowledge graph to enhance recommendations. Furthermore, a soft-matching KGE-based method is proposed to generate interpretable personalized recommendation lists for users.KGAT [19]: This model studies the utility of knowledge graphs, which breaks the independent interactions assumption by associating items with their attributes, and builds an end-to-end graph attention neural network approach to model higher-order connections in KG.DGCF [43]: Disentangled graph collaborative filtering (DGCF) models enhance user intent to disentangle these factors and yield disentangled representations. The specific method is to split the user’s embedding into several segments, each segment representing a specific intent of a user.

### 4.3. Evaluation Metrics

To better examine that the proposed model works under real-world datasets, we evaluate the model by Precision@N, Recall@N, F1@N, and NDCG@N. ${\hat{R}}_{N}$ is a list of the top@N predicted for target user *u*, *N* is the length of the recommendation list from candidate items, and *R* is the test set.

Precision@N: Indicates the proportion of the predicted positive samples that are positive samples.
(19)$$\mathrm{Precision}@N=\frac{|R⋂{\hat{R}}_{N}|}{N}$$

Recall@N: Indicates that the predicted result is the proportion of the actual positive samples in the positive samples to the positive samples in the full sample.
(20)$$\mathrm{Recall}@N=\frac{|R⋂{\hat{R}}_{N}|}{\left|R\right|}$$

F1@N: The F1 score is a weighted average of precision and recall. The higher the F1 score, the more robust the model.
(21)$$F1@N=2\cdot \frac{\mathrm{Precision}@N\cdot \mathrm{Recall}@N}{\mathrm{Precision}@N+\mathrm{Recall}@N}$$

NDCG (Normalized Discounted Cumulative Gain): The score of NDCG represents the correlation between the recommendation list and the target user.
(22)$$\mathrm{NDCG}@N=n_{u}\frac{\sum_{j}2^{r(z)}}{log(1+z)}$$
In the above equation, $n_{u}$ is a normalization constant. r(z) is the correlation coefficient of the *z*-th item in the recommended list, which is generally set to an integer.

### 4.4. Experiment Settings

We implement our model in the Pytorch deep-learning framework. The embedding size is fixed to 64 for all models. All models are optimized by the optimizer Adam, which fixes the batch size to 4096. Model parameters are initialized using the default Xavier initializers [61]. We employ a generic grid search to determine model hyperparameters: the learning rate is adjusted in {0.05, 0.01, 0.005, 0.001}, L2 normalization coefficients are searched in {$10^{-5}$, $10^{-4}$, …, 10,
$10^{2}\}$, and the dropout ratio is uniformly fixed to 0.1.

Furthermore, we employ an early-stopping mechanism during the validation process of model training, i.e., if the Recall@20 on the validation set does not increase in 100 consecutive epochs, the training is stopped early. To model the higher-order connectivity, we set the depth L of the model KHGCN to 3 and the hidden layer dimension (64, 32, 16). In Section 4.6.2, the effects of different layer depths *L* on the performance of the model KHGCN are analyzed experimentally. We adopt a bi-interaction strategy for each layer propagation in KHGCN to aggregate the input vectors.

To evaluate the model’s performance for the target user, we take the 10 most recent interactions as the test set and use the rest of the data for training. The model performance is judged by the generated recommendation list and the four evaluation metrics: Precision@20, Recall@20, F1@20, and NDCG@20. We randomly divide each dataset into a training set, validation set, and test set, and average the results using 10-fold cross-validation for our model. Table 2 shows the hyperparameters set in different datasets, where *d* represents the latent dimension, *L* represents the depth of the KHGCN propagation, *K* represents the number of node-unwrapped latent subspaces, *R* represents the number of routing iterations of the graph capsule, and $\lambda_{\Theta }$ represents the weight for L2 regularization.

For fairness, we set the parameters of other comparison algorithms to the same settings as those in Table 2, and other hyperparameters, except those in Table 2, are selected by grid search.

### 4.5. Performance Comparison

To verify the effectiveness of KHGCN for CKG-based recommendation tasks, we compare the results of KHGCN with other contrasting algorithms on three datasets and four evaluation metrics and show the result of the top@20 in Table 3.

Three algorithms are first analyzed without considering the node embeddings: FM, NFM, and BRPMF. BPRMF outperforms FM and NFM because BPRMF benefits from sampling positive and negative samples (triples). Then, we train the model with BPR loss.Three algorithms are first analyzed without considering the node embeddings: neural network methods, such as CKE and CFKG, are significantly better than the above three methods, which verifies the effectiveness of the knowledge-graph embedding in the recommendation system. The results are shown in Table 3, which may reflect the contribution of deep learning methods to some extent.Table 3 shows that our model performs better and is more competitive than CKE and CFKG under the evaluation criteria. It verifies the superiority of the graph neural network structure in the case of graph-structure-based recommendations. To a certain extent, it reflects that our graph neural network model can provide target users with more accurate item recommendations.Compared with the representative graph-based recommendation method KGAT, Table 3 shows that KHGCN outperforms the KGAT method under four evaluation metrics of the three datasets. Specifically, KHGCN achieves a 5.4% average improvement. Because KGAT only builds a simple graph attention mechanism, it also shows the importance of utilizing the decoupling mechanism and the capsule graph network. We attribute the significant improvement to the expressiveness of graph neural networks in modeling multiple capsule propagation layers.DGCF achieves the best performance in the baseline, suggesting that the disentangled structure is beneficial for enhancing representation learning in the recommendation.

Finally, we analyze and summarize the experimental results of the proposed KHGCN. Our model, built with a decoupling mechanism and a capsule graph neural network, yields better performance than all contrasting algorithms. Experimental results validate the ability of KHGCN to model higher-order connections and learn user–item interactions in collaborative knowledge graphs. KHGCN consistently outperforms other comparison algorithms in all indicators in the performance comparison, which verifies the effectiveness of KHGCN enhanced representation learning to a certain extent.

### 4.6. Analysis of Our Model

In this section, to further deepen the understanding of the KHGCN model, we analyze the more essential hyperparameters and components in the KHGCN model. Firstly, we experimentally verify the input dimension of the knowledge graph matching our model. Next, we investigate the sensitivity of KHGCN to the subspace number of disentanglement *K* and the iterations number of graph routing *R*. Third, we examine how changes in different hyperparameters during training affect the model performance. Moreover, we investigate how to select the aggregation mechanism that is most suitable for information dissemination in graph neural networks. Furthermore, we explore the effect of varying the number of layers *L* in graph neural networks on model performance. Lastly, we study how versions are affected by an additional component in our model.

#### 4.6.1. Sensituvity Studies

In this section, we analyze the sensitivity of KHGCN to the subspace number of latent factors *K* = 2, 8 and the number of routing iterations *R* = 1, 2, 4, 5, where our method has the setting *K* = 4 and *R* = 3. As shown in Figure 3 and Table 4, the results show that KHGCN is not very sensitive to the two hyperparameters. Although the model’s performance improves when *K* = 8 compared to when *K* = 4, the computational complexity of the unwrapping part also doubles. Similarly, although the evaluation index Recall@20 on the dataset Yelp2018 has slightly improved vehicle ability, the model requires more routing iterations when *R* = 4.

#### 4.6.2. Effect of Embedding Propagation Layer Numbers

We test if the more propagation layers KHGCN is embedded with, the better the model performance.In particular, we set the embedding propagation layer *L* in [1, 2, 3, 4, 5]. Figure 4 and Table 5 show the performance of our model under different embedding propagation layers. Figure 4 straightforwardly shows that an appropriate increasing of the depth of the model can improve the performance of our model to a certain extent.

Notably, KHGCN-2 and KHGCN-3 significantly outperform KHGCN-1. Experiments show that KHGCN can effectively model higher-order relationships among the nodes in collaborative knowledge graphs, benefiting from the higher-order connectivity of the second- and third-order embedding layers relative to the first-order one. Further stacking too many layers in KHGCN, the performance drops, like with KHGCN-4 and KHGCN-5. The results show that KHGCN may be too profoundly degraded by the influence of noisy nodes embedded in the propagation layer. An excessively deep embedded propagation layer can lead to overfitting and reduce the model’s performance.

#### 4.6.3. Aggregators Analysis

To investigate how different aggregators for ego representations and neighbor representations will affect the performance of KHGCN, we perform experiments for four variants of KHGCN. Note that two single aggregators (GCN and GraphSage) and two Bi-interaction aggregators (Concatenate and Sum) are four variants of KHGCN. Table 6 show that Bi-interaction (Sum) outperforms GCN, GraphSage, and Bi-interaction (Concatenate) on both metrics. We attribute the improvement to feature interactions, which model affinities between the ego representation and neighbor representations. The result justifies the effectiveness and rationality of the Bi-interaction (Sum) aggregator to capture the heterogeneity of these four representations.

Comprehensive ablation studies are carried out in this section to understand the contribution of each component (i.e., disentangled graph capsules, attention layer, and capsule layers) in our method. We perform the ablation experiments on the component of the KHGCN to analyze their influence on the model performance. Here, we would like to examine how each factor contributes to the final performance. The results evaluated by evaluations are shown in Table 7 and Figure 5, Figure 6 and Figure 7. To see this, we prepare five variants for comparison:w/o KGE: We disable the TransR embedding component of KHGCN. The variant removes the KG entities and their links from HKG but keeps the other nodes and links.w/o Att: We disable the attention mechanism and set π(h,r,t) as $1/|N_{h}|$. The variant replaces the TGAT in the primary capsules with an average pool GNN component.w/o K&A: We obtain the variant by removing both components (w/o KGE and w/o Att).w/o Dis: Directly use the input node representation to serve as graph capsules, without considering the disentanglement factors. The variant removes the nodes disentangled from KHGCN but retains the KG entities and their links.w/o Res: remove the residual connection among the adjacent capsule layers.

Table 7 and Figure 5, Figure 6 and Figure 7 show the performance comparison between the complete model and the five variants. We summarize the experimental results in Table 7, and the results illustrated reveal the following:Removing knowledge-graph embedding and attention components degrade the model’s performance. w/o K&A consistently underperforms compared to w/o KGE and w/o Att. It makes sense since w/o K&A fails to explicitly model the representation relatedness on the granularity of triplets. We can see that removing KG data significantly affects the performance of our model, which further verifies the usefulness of KG data.Compared with w/o Att, w/o KGE performs better in most cases. One possible reason is that treating all neighbors equally might introduce noises and mislead the embedding propagation process. It verifies the substantial influence of graph attention mechanisms.The variant w/o Dis removing nodes disentanglement gives a worse result than the complete model, which shows that disentangling node representation allows us to characterize the latent factors underlying each node and, in turn, more accurately preserve the node/graph properties and capture the part–whole relationship.Furthermore, the w/o variant Residual dropping the residual component is worse than the complete model, which indicates that combining fine, low-layer information with coarse, high-layer information gives us the ability to enhance the final graph-level representation.

Thus, we conclude that each component in our method is necessary and contributes to performance improvement.

## 5. Conclusions and Future Work

This work argues that there are complex and valuable relationships in the collaborative knowledge graph, and the knowledge-graph-based recommendation is essential. With this in mind, we propose a recommendation model with a knowledge-graph capsule network that employs a disentanglement mechanism to handle input node embeddings and a hierarchical capsule graph neural network layer to model higher-order connections and enhance representation learning for collaborative knowledge graphs. Extensive comparative experiments on three large-scale real-world datasets validate the rationality and effectiveness of KHGCN for modeling user and item representations in CKGs. Furthermore, an in-depth analysis of KHGCN demonstrates the usefulness and necessity of the individual components that make up the model KHGCN. This work focuses on the accuracy of recommendations, while ignoring other supplemental inaccuracy metrics of recommendations, such as diversity, novelty, coverage, etc. This work only explores information from entities, relationships, and users, while ignoring information other than user–item interactions and knowledge graphs, which may result in inaccurate recommendations. In addition, most of the existing knowledge-aware recommendations focus on the strong connection between entities, but the user’s fine-grained preference for the item is not easy to capture. Next, we plan to introduce more fine-grained recommendations through the introduction of multimodal aware information.

In the future, we will continue to conduct research in the following directions based on the current research results:Diversify input information and increase advanced representation learning models;Furthermore, to build a more comprehensive knowledge graph, we will explore extracting entity-related information from other sources, such as text, images, etc.

## Figures and Tables

**Figure 1 entropy-25-00697-f001:**
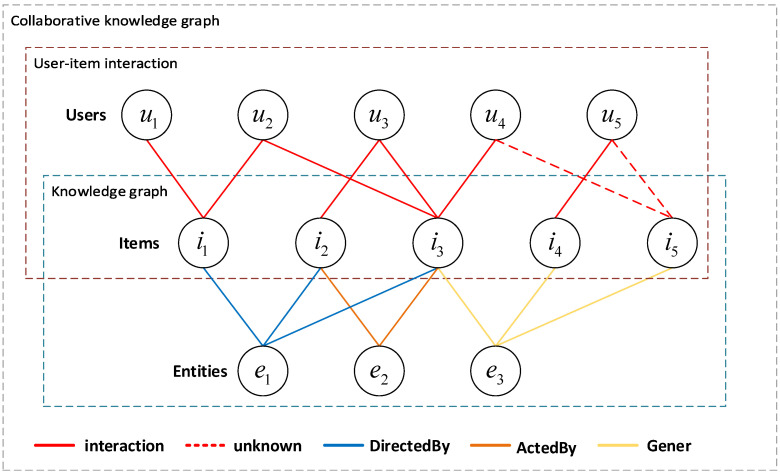
A toy example of the CKG contains users, movies, entities, interactions, unknown, directed by, and gener as relations.

**Figure 2 entropy-25-00697-f002:**
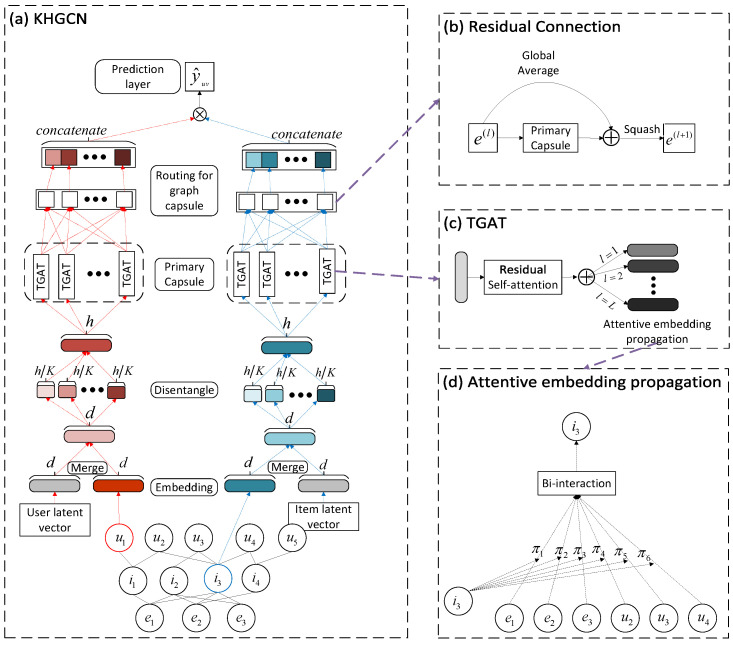
This is a figure. Schemes follow the same formatting. If there are multiple panels, they should be listed as follows: (**a**) The framework of KHGCN. (**b**) The residual connection in the routing part. (**c**) The TGAT layer part in the primary graph capsule. (**d**) The attentive embedding propagation layer part in the TGAT.

**Figure 3 entropy-25-00697-f003:**
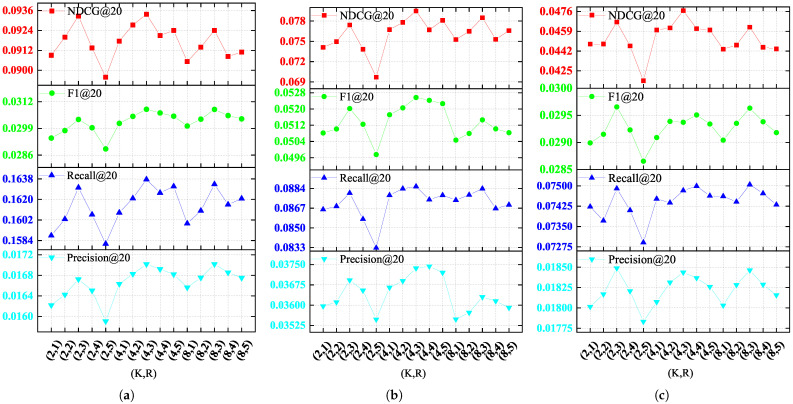
Sensitivity studies for the top@20 recommendation. (**a**) Amazon-Book. (**b**) Last-Fm. (**c**) Yelp2018.

**Figure 4 entropy-25-00697-f004:**
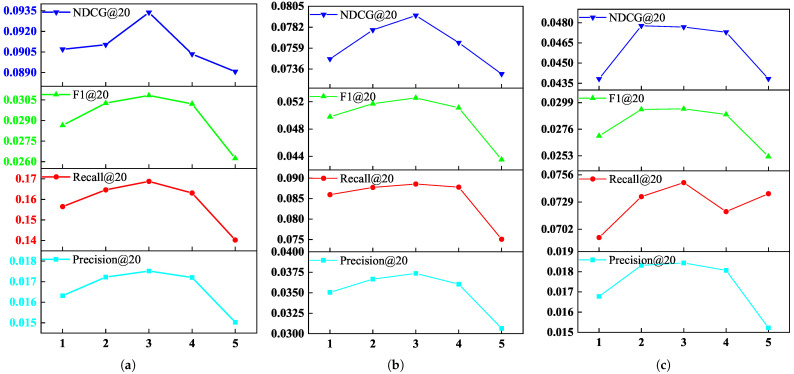
Effect of embedding propagation layer numbers for the top@20 recommendation. (**a**) Amazon-Book. (**b**) Last-Fm. (**c**) Yelp2018.

**Figure 5 entropy-25-00697-f005:**
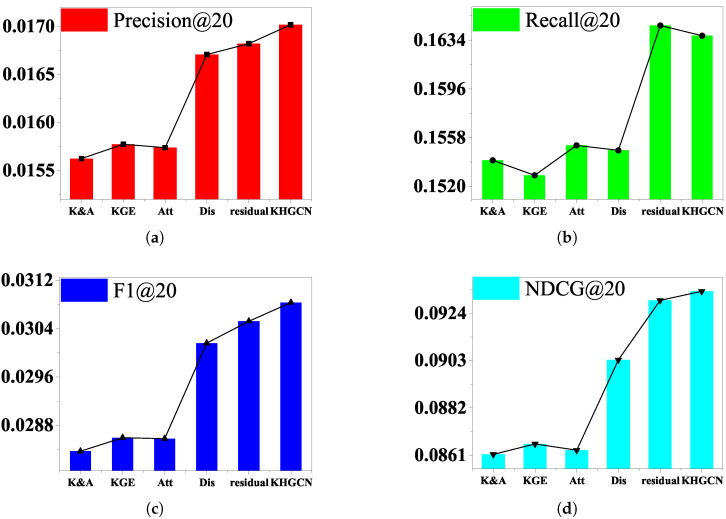
Ablation analysis on Amazon-Book for the top@20 recommendation. (**a**) Precision@20. (**b**) Recall@20. (**c**) F1@20. (**d**) NDCG@20.

**Figure 6 entropy-25-00697-f006:**
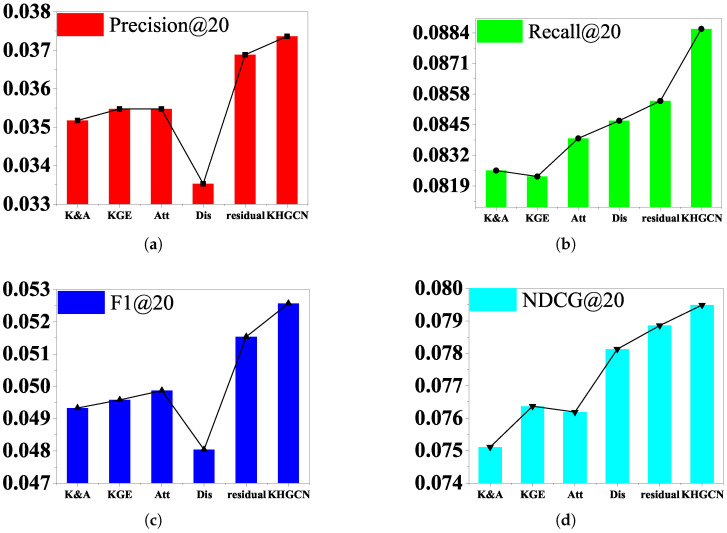
Ablation analysis on Last-Fm for the top@20 recommendation. (**a**) Precision@20. (**b**) Recall@20. (**c**) F1@20. (**d**) NDCG@20.

**Figure 7 entropy-25-00697-f007:**
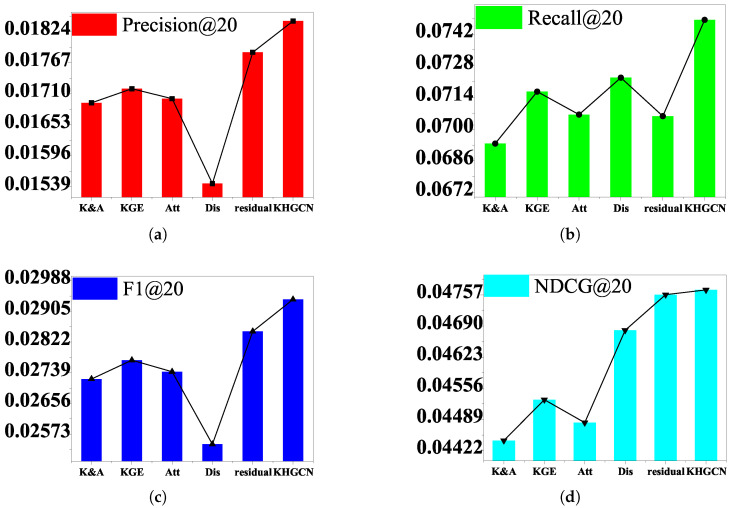
Ablation analysis on Yelp2018 for the top@20 recommendation. (**a**) Precision@20. (**b**) Recall@20. (**c**) F1@20. (**d**) NDCG@20.

**Table 1 entropy-25-00697-t001:** Details of the three datasets.

Datasets		Amazon-Book	Last-FM	Yelp2018
User–Item Interaction	#Users	70,679	23,566	45,919
	#Items	24,915	48,123	45,538
	#Interactions	847,733	3,034,796	1,185,068
	#Density	0.000481	0.002676	0.0005667
Knowledge Graph	#Entities	88,572	58,266	90,961
	#Relations	39	9	42
	#Triplets	2,557,746	464,567	1,853,704

**Table 2 entropy-25-00697-t002:** Hyperparameters.

Datasets	Amazon-Book	Last-FM	Yelp2018
*d*	64	64	64
*L*	3	3	3
*K*	4	4	4
*R*	3	3	3
$\lambda_{\Theta }$	0.0001	0.0001	0.00001

**Table 3 entropy-25-00697-t003:** Performance of different methods for the top@20 recommendation.

	Amazon-Book					Last-FM					Yelp2018			
	Precision	Recall	F1	NDCG		Precision	Recall	F1	NDCG		Precision	Recall	F1	NDCG
FM	0.0144	0.1449	0.0262	0.0722		0.0306	0.0791	0.0442	0.0647		0.0157	0.0657	0.0253	0.0415
NFM	0.0139	0.1383	0.0252	0.0725		0.0293	0.0768	0.0424	0.0610		0.0144	0.0623	0.0235	0.0397
BPRMF	0.0142	0.1336	0.0256	0.0694		0.0316	0.0757	0.0446	0.0649		0.0163	0.0670	0.0262	0.0432
ECFKG	0.0130	0.1225	0.0236	0.0612		0.0293	0.0744	0.0420	0.0617		0.0147	0.0600	0.0236	0.0386
CKE	0.0145	0.1384	0.0263	0.0723		0.0315	0.0756	0.0445	0.0648		0.0165	0.0676	0.0265	0.0438
KGAT	0.0149	0.1416	0.0270	0.0756		0.0329	0.0802	0.0466	0.0688		0.0163	0.0670	0.0262	0.0429
DGCF	0.0151	0.1524	0.0277	0.0756		0.0320	0.0825	0.0463	0.0682		0.0164	0.0693	0.0263	0.0431
KHGCN	0.0170	0.1638	0.0308	0.0934		0.0374	0.0886	0.0526	0.0795		0.0184	0.0748	0.0294	0.0477

**Table 4 entropy-25-00697-t004:** Sensitivity studies for the top@20 recommendation.

		Amazon-Book				Last-FM					Yelp2018				
		**Precision**	**Recall**	**F1**	**NDCG**		**Precision**	**Recall**	**F1**	**NDCG**		**Precision**	**Recall**	**F1**	**NDCG**
K	2	0.0167	0.1630	0.0303	0.0933		0.0370	0.0880	0.0520	0.0774		0.0185	0.0749	0.0297	0.0467
	8	0.0170	0.1633	0.0308	0.0924		0.0363	0.0884	0.0515	0.0785		0.0185	0.0750	0.0296	0.0463
R	1	0.0166	0.1608	0.0301	0.0918		0.0367	0.0878	0.0517	0.0767		0.0181	0.0745	0.0291	0.0460
	2	0.0168	0.1621	0.0304	0.0927		0.0369	0.0884	0.0521	0.0778		0.0183	0.0744	0.0294	0.0462
	4	0.0169	0.1626	0.0307	0.0921		0.0374	0.0874	0.0524	0.0767		0.0184	0.0750	0.0295	0.0461
	5	0.0168	0.1631	0.0305	0.0924		0.0372	0.0878	0.0523	0.0781		0.0183	0.0746	0.0293	0.0460
*K* = 4, *R* = 3		0.0170	0.1638	0.0308	0.0934		0.0374	0.0886	0.0526	0.0795		0.0184	0.0748	0.0294	0.0477

**Table 5 entropy-25-00697-t005:** Effect of embedding propagation layer numbers for the top@20 recommendation.

	Amazon-Book					Last-FM					Yelp2018			
	Precision	Recall	F1	NDCG		Precision	Recall	F1	NDCG		Precision	Recall	F1	NDCG
*L* = 1	0.0158	0.1515	0.0286	0.0907		0.0350	0.0860	0.0498	0.0747		0.0168	0.0694	0.0270	0.0438
*L* = 2	0.0167	0.1596	0.0303	0.0910		0.0367	0.0877	0.0517	0.0779		0.0183	0.0734	0.0293	0.0478
*L* = 3	0.0170	0.1638	0.03088	0.0934		0.0374	0.0886	0.0526	0.0795		0.0184	0.0748	0.0294	0.0477
*L* = 4	0.0167	0.1581	0.0302	0.0903		0.0361	0.0878	0.0511	0.0765		0.0181	0.0720	0.0289	0.0473
*L* = 5	0.0145	0.1352	0.0262	0.0891		0.0306	0.0751	0.0435	0.0731		0.0152	0.0737	0.0252	0.0438

**Table 6 entropy-25-00697-t006:** Aggregators analysis for the top@20 recommendation.

Aggregators	Amazon-Book					Last-FM					Yelp2018			
	Precision	Recall	F1	NDCG		Precision	Recall	F1	NDCG		Precision	Recall	F1	NDCG
GCN	0.0165	0.1568	0.0298	0.0904		0.0361	0.0857	0.0508	0.07697		0.0177	0.0720	0.0285	0.0461
GraphSage	0.0168	0.1605	0.0304	0.0921		0.0359	0.0868	0.0508	0.07707		0.0178	0.0734	0.0286	0.0462
Concatenate	0.0167	0.1615	0.0303	0.0918		0.0367	0.0872	0.0517	0.0782		0.0181	0.0735	0.0291	0.0469
Sum	0.0170	0.1638	0.0308	0.0934		0.0374	0.0886	0.0526	0.0795		0.0184	0.0748	0.0294	0.0477

**Table 7 entropy-25-00697-t007:** Ablation analysis for the top@20 recommendation.

	Amazon-Book					Last-FM					Yelp2018			
	Precision	Recall	F1	NDCG		Precision	Recall	F1	NDCG		Precision	Recall	F1	NDCG
w/o K&A	0.0156	0.1540	0.0283	0.0861		0.0352	0.0826	0.0493	0.0751		0.0169	0.0693	0.0272	0.0444
w/o KGE	0.0158	0.1529	0.0286	0.0866		0.0355	0.0823	0.0496	0.0764		0.0172	0.0717	0.0277	0.0453
w/o Att	0.0157	0.1552	0.0286	0.0863		0.0355	0.0840	0.0499	0.0762		0.0170	0.0706	0.0274	0.0448
w/o Dis	0.0167	0.1548	0.0302	0.0903		0.0335	0.0847	0.0480	0.0781		0.0155	0.0723	0.0255	0.0468
w/o Res	0.0168	0.1645	0.0305	0.0930		0.0369	0.0855	0.0515	0.0789		0.0179	0.0706	0.0285	0.0476
KHGCN	0.0170	0.1638	0.0308	0.0934		0.0374	0.0886	0.0526	0.0795		0.0184	0.0748	0.0294	0.0477

## Data Availability

Data will be made available on request.

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
