# Peer review of "KHGCN: Knowledge-Enhanced Recommendation with Hierarchical Graph Capsule Network"

_entropy, 2023, doi:10.3390/e25040697_

Round 1

Reviewer 1 Report

1) Ontologies, as a potential base of knowledge graph, contains reviewed/verified knowledge in hierarchical structure. I am just wondering why there is no discussion regarding ontology, since hierarchical structure, as well as noise introduced by hierarchies are main topics of this article. Could you please discuss the hierarchical structure of ontologies, and if your method could help solve the problems accordingly for ontology-based knowledge inference (parent-child relationship between classes, class & individuals, properties?)

2) Currently ontologies always have complex hierarchies (e.g., more then 10 levels hierarchies). How could your method help deal with it? 

Author Response

Response to Reviewer 1 Comments

Point 1: Ontologies, as a potential base of knowledge graph, contains reviewed/verified knowledge in hierarchical structure. I am just wondering why there is no discussion regarding ontology, since hierarchical structure, as well as noise introduced by hierarchies are main topics of this article.

Response 1: Thanks very much for your suggestion. Recommendation based on knowledge graph studies the application of knowledge graph, which is carried out on the basis of established knowledge base and only involves specific entities (i.e., ontology + instance). Therefore, most recommendation studies based on knowledge graph (e.g., DGCF, KGAT, etc.) don’t discuss ontology.

Point 2: Could you please discuss the hierarchical structure of ontologies, and if your method could help solve the problems accordingly for ontology-based knowledge inference (parent-child relationship between classes, class & individuals, properties?).

Response 2: Thanks very much for your suggestion. The model proposed by us is also a fine-grained learning of the embedded representation of nodes and relations in the knowledge graph. Our method can also be used as the representation learning part of knowledge reasoning, so it is helpful for ontology-based knowledge inference problems.

Point 3: Currently ontologies always have complex hierarchies (e.g., more than 10 levels hierarchies). How could your method help deal with it?

Response 3: Thanks very much for your suggestion. In the knowledge-based recommendation, the complex hierarchies (e.g., more than 10 levels hierarchies) have not been studied. For example, DGCF and KGAT methods studied 4 levels hierarchies. Due to sparse data in the recommendation system, the excessively complex hierarchy structure introduces too much noise. Meanwhile, the excessively complex hierarchy structure involves the construction of large-scale graph and the calculation of cost, and there are still no good solutions to these problems.

Reviewer 2 Report

This paper proposes a recommendation model for collaborative knowledge graphs (CKGs) using a knowledge graph capsule network (KHGCN). The paper argues that there are valuable relationships within CKGs, and knowledge graph-based recommendations are crucial for effective recommendation systems. The KHGCN model employs a disentanglement mechanism to handle input node embeddings and a hierarchical capsule graph neural network layer to model higher-order connections and enhance representation learning. The paper presents extensive comparative experiments on three large-scale real-world datasets to validate the effectiveness of KHGCN in modeling user and item representations in CKGs. Additionally, an in-depth analysis of KHGCN clearly demonstrates the usefulness and necessity of the individual components of the model.

However, the paper focuses solely on the accuracy of recommendations, which indeed return good performance, but it ignores other supplemental metrics such as diversity and coverage. Furthermore, the model only explores information from entities, relationships, and users, which may result in limited recommendation performance. The authors also mentioned that existing knowledge-aware recommendations may not be able to capture the user's fine-grained preference for the item.

Overall, the paper provides valuable insights into the use of CKGs for recommendation systems and proposes a promising model for representation learning in CKGs. 

One minor suggestion is to include Section 3 into Section 4, one whole section for a graph without any detail is confusing.

Author Response

Response to Reviewer 2 Comments

Point 1:This paper proposes a recommendation model for collaborative knowledge graphs (CKGs) using a knowledge graph capsule network (KHGCN). The paper argues that there are valuable relationships within CKGs, and knowledge graph-based recommendations are crucial for effective recommendation systems. The KHGCN model employs a disentanglement mechanism to handle input node embeddings and a hierarchical capsule graph neural network layer to model higher-order connections and enhance representation learning. The paper presents extensive comparative experiments on three large-scale real-world datasets to validate the effectiveness of KHGCN in modeling user and item representations in CKGs. Additionally, an in-depth analysis of KHGCN clearly demonstrates the usefulness and necessity of the individual components of the model.

However, the paper focuses solely on the accuracy of recommendations, which indeed return good performance, but it ignores other supplemental metrics such as diversity and coverage.

Response 1: Thank you very much for your valuable suggestion. In the knowledge-based recommendation, due to the sparse data, it is not a good choice to directly use coverage rate as an evaluation index. Therefore, most recommendation algorithms in Top@K use Recall as a replacement index of coverage. Recall reflects the comprehensiveness of recommendation, Recall and Precision are mutually inverse, and Recall and Precision are contradictory measures. The higher the Precision, the lower the Recall, and vice versa. So both the accuracy and comprehensiveness of recommendations are considered.

Point 2: Furthermore, the model only explores information from entities, relationships, and users, which may result in limited recommendation performance.

Response 2: Thank you very much for your valuable comment. The model proposed by us is a fine-grained learning of the embedded representation of nodes and relations in the knowledge graph. Firstly, the nodes are dissociated, and then the whole graph is embedded and weighted by hierarchical capsule diagrams. This allows fine-grained preference modeling of both entities and relationships in the graph, rather than simply weighting the representation of relationships or attributes.

Point 3: The authors also mentioned that existing knowledge-aware recommendations may not be able to capture the user's fine-grained preference for the item.

Response 3: Thank you very much for your valuable comment. Knowledge-based recommendation, which has certain limitations. We are conducting research on multi-modal recommendation, which mainly focuses on integrating text information into knowledge-based recommendation.

Point 4: Overall, the paper provides valuable insights into the use of CKGs for recommendation systems and proposes a promising model for representation learning in CKGs. 

One minor suggestion is to include Section 3 into Section 4, one whole section for a graph without any detail is confusing.

Response 4: Thanks very much for your recognition of our work. We deleted the duplicated Section and improved the quality of our submission seriously.

Round 2

Reviewer 1 Report

n/a